# Controlling fertilization and cAMP signaling in sperm by optogenetics

**Vera Jansen[1,2], Luis Alvarez[1], Melanie Balbach[1], Timo Strünker[1], Peter Hegemann[3], U Benjamin Kaupp[1]\*, Dagmar Wachten[2]\***

[1]Department of Molecular Sensory Systems, Center of Advanced European Studies and Research, Bonn, Germany; [2]Minerva Research Group Molecular Physiology, Center of Advanced European Studies and Research, Bonn, Germany; [3]Institute of Biology, Experimental Biophysics, Humboldt University of Berlin, Berlin, Germany

**Abstract** Optogenetics is a powerful technique to control cellular activity by light. The light-gated Channelrhodopsin has been widely used to study and manipulate neuronal activity in vivo, whereas optogenetic control of second messengers in vivo has not been examined in depth. In this study, we present a transgenic mouse model expressing a photoactivated adenylyl cyclase (bPAC) in sperm. In transgenic sperm, bPAC mimics the action of the endogenous soluble adenylyl cyclase (SACY) that is required for motility and fertilization: light-stimulation rapidly elevates cAMP, accelerates the flagellar beat, and, thereby, changes swimming behavior of sperm. Furthermore, bPAC replaces endogenous adenylyl cyclase activity. In mutant sperm lacking the bicarbonate-stimulated SACY activity, bPAC restored motility after light-stimulation and, thereby, enabled sperm to fertilize oocytes in vitro. We show that optogenetic control of cAMP in vivo allows to non-invasively study cAMP signaling, to control behaviors of single cells, and to restore a fundamental biological process such as fertilization.

**\*For correspondence:**
U.B.Kaupp@caesar.de (UBK);
dagmar.wachten@caesar.de (DW)

**Competing interests:** The authors declare that no competing interests exist.

**Reviewing editor**: David E Clapham, Howard Hughes Medical Institute, Boston Children's Hospital, United States

## Introduction

Almost every eukaryotic cell contains a specialized surface protrusion called the primary cilium (*Singla and Reiter, 2006*). Primary cilia serve as sensory antennae that register physical and chemical cues from the environment. Cilia are not only involved in for example, mechano-, chemo-, and photo-sensation but also in embryonic and neuronal development (*Praetorius and Spring 2001*; *Singla and Reiter, 2006*; *Insinna and Besharse, 2008*; *Goetz and Anderson, 2010*; *Satir et al., 2010*). Motile cilia, also called flagella, are used both as sensory antenna and motors that move fluids or propel cells (*Salathe, 2007*; *Bloodgood, 2010*; *Lindemann and Lesich, 2010*; *Pichlo et al., 2014*). Signaling in cilia and flagella is tightly regulated, spatially confined, and relies on a cilia-specific transport machinery (*Rosenbaum and Witman, 2002*; *Delling et al., 2013*; *Chung et al., 2014*; *Pichlo et al., 2014*).

Cyclic nucleotide-dependent signaling plays a central role in both primary and motile cilia. In ciliary structures of olfactory neurons and photoreceptors, it regulates chemosensation and photoreception, respectively (*Johnson and Leroux, 2010*). In cilia of endothelial and mesenchymal cells, cAMP signaling controls cilia length in response to extracellular stimuli (*Besschetnova et al., 2010*). Defects in cilia of the renal epithelium cause polycystic kidney disease (*Pazour et al., 2000*). During embryonic development, ciliary cAMP signaling is involved in neural tube development by regulating the Sonic hedgehog (Shh) pathway (*Mukhopadhyay et al., 2013*).

A case in point for the importance of cAMP signaling in flagella is the mammalian sperm cell. Cyclic AMP signaling is essential for sperm development, motility, and maturation in the female genital tract (*Visconti et al., 1995b*; *Wennemuth et al., 2003*; *Krähling et al., 2013*). However, the underlying signaling pathways are not well understood. CRIS, a cAMP-binding protein, controls spermatogenesis

**eLife digest** Tiny hair-like structures called cilia on the outside of cells play many important roles, including detecting physical and chemical signals from the environment. Special cilia—called flagella—help cells to move around and perhaps the most well-known of these are sperm flagella, which propel sperm in their quest to fertilize the egg. A chemical messenger called cAMP is essential for the movement of sperm flagella.

When a sperm cell enters the female reproductive tract, an enzyme called SACY is activated. Within seconds, SACY produces cAMP and, thereby, causes the flagella to beat faster so that the sperm cell speeds toward the egg. cAMP also controls sperm maturation, which is needed to penetrate the egg. However, the precise details of the role of cAMP in sperm cells are not clear.

Here, Jansen et al. have investigated this role using a cutting-edge technique—called optogenetics—that was originally developed to study brain cells in living organisms. Jansen et al. genetically engineered a mouse so that exposing sperm to blue light activates a light-sensitive enzyme called bPAC that increases cAMP levels in sperm.

In these mice, the activation of bPAC by light accelerated the beating of the flagella so the sperm moved faster, in a way that was similar to the effects that are normally observed after the activation of the SACY enzyme. In mice lacking among other things the SACY enzyme—whose sperm cells are unable to move or fertilize an egg—activating the light-sensitive bPAC enzyme restored sperm motility and enabled the sperm to fertilize an egg.

These results show that optogenetics may be a useful tool for studying how flagella and other types of cilia work.

and sperm motility (*Krähling et al., 2013*). An increase in bicarbonate ($HCO_3^-$) during the transit from the epididymis to the female genital tract activates a soluble adenylyl cyclase (SACY) and, thereby, accelerates the flagellar beat and enhances progressive sperm motility (*Esposito et al., 2004*; *Xie et al., 2006*). $HCO_3^-$-induced cAMP synthesis by SACY stimulates protein kinase A (PKA) (*Wennemuth et al., 2003*; *Nolan et al., 2004*) and controls capacitation, a maturation process of sperm in the female genital tract that is essential for fertilization (*Chang, 1951*; *Austin, 1952*). Acceleration of the flagellar beat and capacitation proceed on quite different time scales: the beat frequency increases within seconds, whereas PKA-dependent tyrosine phosphorylation, a hallmark of sperm capacitation, takes about an hour to become noticeable (*Visconti et al., 1995a*).

The study of cAMP-dependent signaling pathways in mammalian sperm is further complicated by an ill-defined interplay between cAMP and $Ca^{2+}$ signaling; for example, $Ca^{2+}$ controls SACY activity (*Carlson et al., 2007*) and cAMP reportedly evokes a $Ca^{2+}$ influx (*Kobori et al., 2000*; *Ren et al., 2001*; *Xia et al., 2007*; *Wertheimer et al., 2013*). Previous attempts to delineate these signaling pathways relied on common pharmacological tools: membrane-permeable cAMP analogs and modulators of cAMP signaling components (*Kobori et al., 2000*; *Xia et al., 2007*; *Wertheimer et al., 2013*). Recent studies in human sperm, however, refute that cAMP controls $[Ca^{2+}]_i$ (*Strünker et al., 2011*) and disclose serious shortcomings of these pharmacological tools that directly act on the sperm $Ca^{2+}$ channel CatSper and, thereby, artifactually change $Ca^{2+}$ levels (*Brenker et al., 2012*). In mice, however, it is equivocal whether cAMP indeed evokes a $Ca^{2+}$ signal. Therefore, it is required to study the interconnection of cAMP and $Ca^{2+}$ signaling in mouse sperm by novel, non-invasive techniques.

Optogenetics is a powerful tool to spatially and temporally control second messenger-dependent signaling, uncompromised by pharmacological side effects. To this end, we present here a transgenic mouse model expressing the photoactivated adenylyl cyclase bPAC (*Ryu et al., 2010*; *Stierl et al., 2011*) to manipulate cAMP levels in sperm and, thereby, control sperm motility by light. We show that bPAC mimics the action of $HCO_3^-$ in sperm. Furthermore, bPAC functionally replaces the endogenous SACY activity and remedies cAMP signaling defects, demonstrating that in vitro, fertility can be restored by optogenetics. Finally, an increase of cAMP levels by light does not elevate $Ca^{2+}$ levels in sperm. Thus, the Prm1-bPAC mouse model is a powerful tool to analyze cAMP-dependent signaling in ciliary structures with high spatio-temporal resolution uncompromised by pharmacological artifacts.

## Results

### Generation of Prm1-bPAC transgenic mice

We engineered a targeting vector to express the beta subunit of photo activated adenylyl cyclase from the soil bacterium *Beggiatoa* (bPAC) under the control of the protamine 1 promoter (Prm1, *Figure 1A*) that is exclusively active in post-meiotic spermatids (*Zambrowicz et al., 1993*). Transgenic mice were generated by pronuclear injection using standard procedures (*Ittner and Götz, 2007*). Genomic insertion of the transgene was confirmed by PCR (*Figure 1B*). Protein expression of bPAC in testis lysates varied between different founder lines (*Figure 1C*). This variation in transgene expression reflects differences in integration site and/or copy number. For further analysis, founder lines 1 and 5 were chosen due to stable inheritance of the transgene to the next generation. The bPAC protein was exclusively expressed in sperm (*Figure 1D–F*). Prm1-bPAC males were fertile and did not show any defects during spermatogenesis, demonstrating that bPAC expression does not affect sperm development or function (*Figure 1—source data 1*).

### Control of sperm cAMP levels and motility by light

To scrutinize whether bPAC allows optogenetic control of sperm cAMP signaling, we determined cAMP levels in whole sperm before and after light stimulation. Basal cAMP levels in bPAC sperm were slightly enhanced (bPAC: $15.3 \pm 3.6$ fmol/$10^5$ sperm versus wild-type: $10.0 \pm 4.4$ fmol/$10^5$ sperm; *Figure 2A*), which might be due to basal bPAC activity (*Stierl et al., 2011*). In wild-type sperm, light stimulation did not alter cAMP levels, whereas in bPAC sperm, cAMP levels increased by about 2.5-fold (*Figure 2A*). During continuous light stimulation, cAMP levels reached a maximum within 2 min (*Figure 2B*); after switching off the light, cAMP returned to baseline within 10 min (*Figure 2B*).

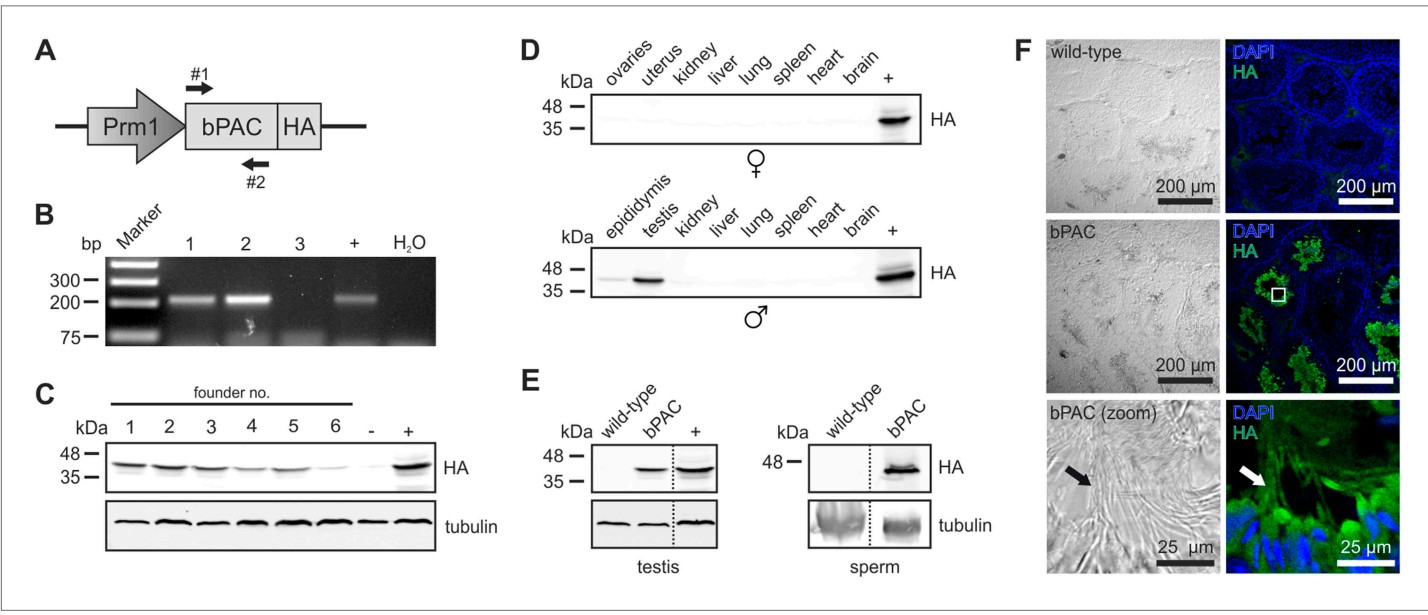

**Figure 1**. Characterization of the Prm1-bPAC mouse. (**A**) Scheme of the Prm1-bPAC targeting vector. Expression of hemagglutinin (HA)-tagged bPAC is driven by the protamine 1 promoter (Prm1); arrows indicate the position of genotyping primers. (**B**) Genotyping by PCR. In Prm1-bPAC mice, a 213-bp fragment is amplified. The targeting vector served as a positive control (+). (**C**) Western blot analyzing bPAC-HA expression in testis lysates from different founder lines. Lysates from HEK cells expressing bPAC-HA served as positive control (+), wild-type testis lysates as negative control (−). (**D**) Western blot analyzing bPAC-HA expression in tissue lysates from male and female Prm1-bPAC mice. (**E**) Western blot analyzing bPAC-HA expression in testis and sperm. (**F**) Immunohistochemical analysis of bPAC-HA expression (left panel: transmission, right panel: fluorescence). Pictures at the bottom show a higher magnification (see white box). Sperm flagella are indicated (arrow). Cryosections of mouse testis were probed with anti-HA antibody and fluorescent secondary antibody (green), DNA was stained with DAPI (blue). Loading control for Western blots: β-tubulin.

The following source data is available for figure 1:

**Source data 1**. The Prm1-bPAC mouse model shows no change in fertility parameters.

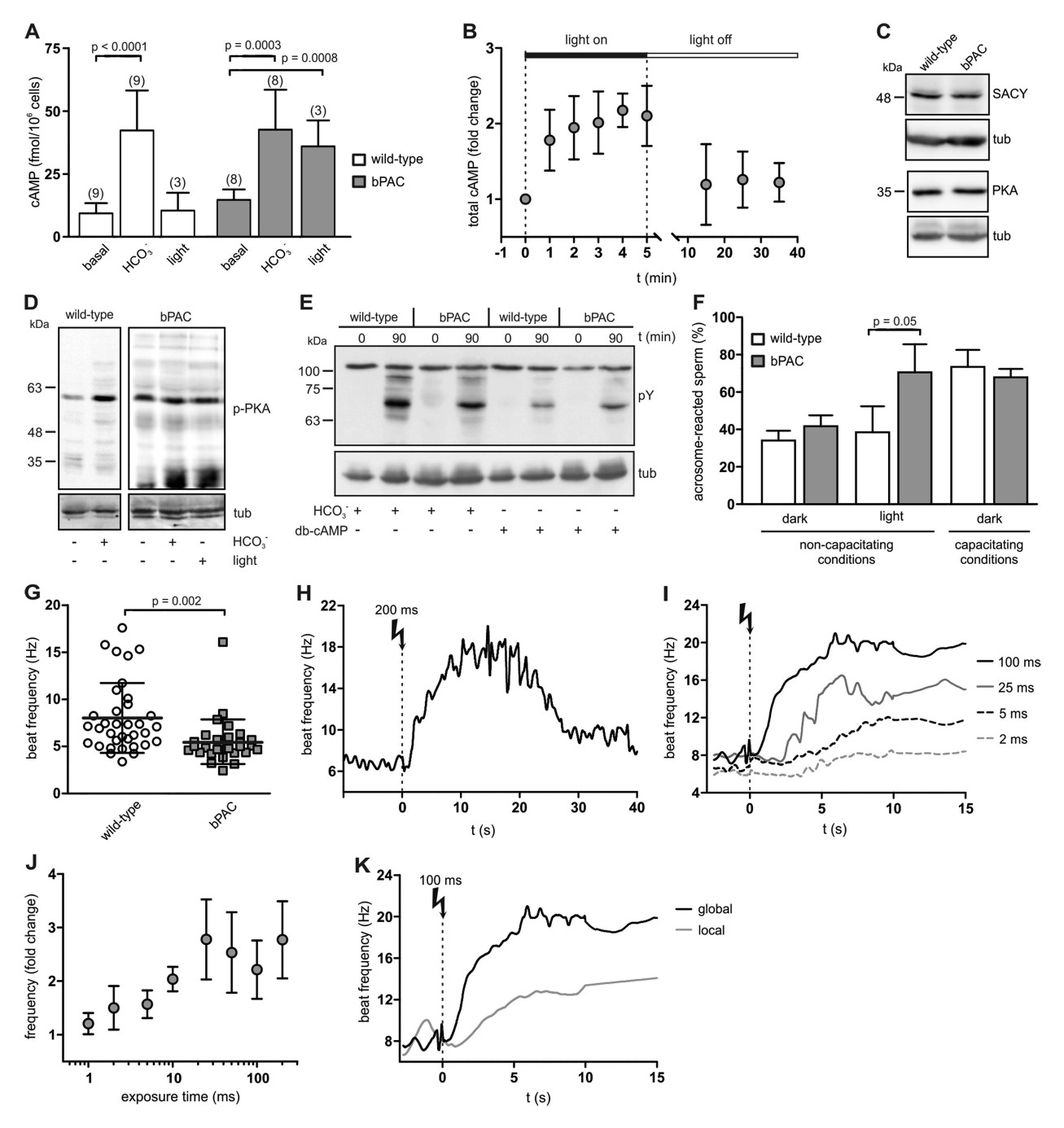

**Figure 2**. Manipulation of cAMP signaling and sperm motility by light. (**A**) Light stimulation of bPAC sperm for 10 min increases cAMP levels. Stimulation of SACY activity with $HCO_3^-$ (1 min) evokes the same response in wild-type and bPAC sperm. (**B**) cAMP levels in bPAC sperm during prolonged light stimulation and after switching off the light. (**C**) Western blot analyzing SACY and PKA expression in wild-type and bPAC sperm. (**D**) Phosphorylation of PKA targets in wild-type and bPAC sperm detected with an anti-phospho-(Ser/Thr) PKA substrate antibody (p-PKA). Wild-type sperm were stimulated with 25 mM $HCO_3^-$, bPAC sperm with $HCO_3^-$ or light. (**E**) db-cAMP- (1 mM) and $HCO_3^-$-induced tyrosine phosphorylation in wild-type and bPAC sperm detected with an anti-phospho-tyrosine antibody (pY). (**F**) Acrosome-reaction assay. Percentage of sperm that has undergone the acrosome reaction under non-capacitating conditions (no $HCO_3^-$) before and after light-stimulation and under capacitating conditions (25 mM $HCO_3^-$). (**G**) Basal flagellar beat frequency in wild-type and bPAC sperm (individual values and mean ± s.d.). (**H**, **I**) Light-induced change in flagellar beat frequency of individual bPAC sperm. (**J**) Average change in flagellar beat frequency after light stimulation. (**K**) Change in flagellar beat frequency of bPAC sperm after global or local light stimulation of the flagellum for 100 ms; see also *Video 3 and 4*. Data are plotted as mean ± s.d.; (n) = number of experiments, n ≥ 5 if not stated otherwise; p values calculated using Student's t test. Loading control for Western blots: β-tubulin.

The light-induced cAMP increase was similar to the $HCO_3^-$-induced cAMP increase (*Figure 2A*), indicating that activation of bPAC mimics the activation of SACY by $HCO_3^-$.

Next, we studied whether bPAC expression interferes with cellular events controlled by cAMP, such as capacitation. Sperm are capacitated in vitro by incubation with a cholesterol acceptor (e.g., serum albumin) and $HCO_3^-$ to stimulate cAMP synthesis via SACY (*Visconti et al., 1995a*; *Xie et al., 2006*). cAMP in turn activates PKA, the principal cAMP target in sperm. PKA activation indirectly promotes tyrosine phosphorylation of a subset of proteins, which is considered to be a hallmark of capacitation (*Visconti et al., 1995a*; *Nolan et al., 2004*).

The presence of bPAC did not affect SACY expression level (*Figure 2C*), consistent with the observation that $HCO_3^-$-stimulated cAMP synthesis is similar in wild-type and bPAC sperm (*Figure 2A*). The expression level of PKA also remained unchanged in bPAC sperm (*Figure 2C*). The phosphorylation of PKA targets under non-stimulated conditions was also not majorly different in bPAC sperm compared to wild-type sperm (*Figure 2D*). Furthermore, stimulation of bPAC sperm with either light or $HCO_3^-$ both evoked phosphorylation of PKA targets (*Figure 2D*). Finally, we analyzed whether the capacitation-associated phosphorylation of tyrosine residues was altered: incubation with $HCO_3^-$ or the cAMP analog dibutyryl-cAMP (db-cAMP) enhanced tyrosine phosphorylation to a similar extent in wild-type and bPAC sperm (*Figure 2E*). In summary, bPAC expression does not interfere with known cAMP-signaling events in sperm.

Next, we studied whether a cAMP increase evoked by light-stimulation of bPAC results in sperm capacitation. As a functional read-out for sperm capacitation, we used an acrosome-reaction assay (*Figure 2F*). Under non-capacitating conditions (without $HCO_3^-$) in the dark, the percentage of sperm undergoing the acrosome reaction was similar between wild-type and bPAC sperm (wild-type: 34.1% versus bPAC: 41.7%; *Figure 2F*). After light-stimulation, the number of sperm undergoing the acrosome reaction changed dramatically for bPAC sperm but remained unchanged for wild-type sperm (wild-type: 38.4% bPAC: 70.5%; *Figure 2F*). This demonstrates that a light-stimulated increase in cAMP levels results in sperm capacitation and that bPAC functionally mimics the activation of SACY by $HCO_3^-$.

In mammalian sperm, cAMP controls the flagellar beat (*Wennemuth et al., 2003*; *Esposito et al., 2004*; *Nolan et al., 2004*). The basal beat frequency of bPAC sperm was slightly lower compared to wild-type sperm (5.5 ± 2.5 Hz versus 8.0 ± 3.7 Hz; *Figure 2G*), which might be due to a cAMP-dependent feedback mechanism activated by elevated basal cAMP levels (*Burton and McKnight, 2007*). Light stimulation of bPAC sperm evoked a transient acceleration of the flagellar beat: a 200-ms light pulse increased the beat frequency within 10 s about 2.5-fold from 7 Hz to 18 Hz (*Figure 2H*; *Videos 1,2*). The acceleration persisted for at least 10 s and sperm returned to basal beat frequency within 30 s (*Figure 2H*). The light-stimulated acceleration of the flagellar beat was dose-dependent: the longer the light exposure, the faster the onset and higher the beat frequency (*Figure 2I,J*). Local illumination of a small part of the flagellum also enhanced the beat frequency, albeit the increase was smaller and its kinetics slower compared to global illumination (*Figure 2K*, *Videos 3,4*). Altogether, we conclude that (1) light activation of bPAC mimics the transient beat acceleration evoked by $HCO_3^-$ stimulation of SACY (*Wennemuth et al., 2003*; *Carlson et al., 2007*), (2) the behavioral response of single cells can be modulated in a graded fashion, and that (3) bPAC is able to control cAMP signaling in the sperm flagellum with spatial precision.

## Light-stimulated cAMP synthesis does not affect intracellular $Ca^{2+}$ levels

We tested whether a light-induced increase of cAMP evokes a $Ca^{2+}$ influx into bPAC sperm. The use of a red-shifted fluorescent $Ca^{2+}$ indicator enabled us to orthogonalize indicator excitation and bPAC activation (*Stierl et al., 2011*): after recording basal $Ca^{2+}$ levels at 520 nm, we repeatedly switched between excitation of the $Ca^{2+}$ indicator at 520 nm and activation of bPAC at 485 nm; finally, we challenged sperm with 8-Br-cNMPs or db-cAMP while exciting the $Ca^{2+}$ indicator only (*Figure 3A–C*). To scrutinize our experimental conditions, we determined sperm cAMP levels before and after activation of bPAC (*Figure 3D*). We observed that cAMP synthesis via light stimulation of bPAC did not evoke a $Ca^{2+}$ signal (*Figure 3A–D*), consistent with the observation that a $HCO_3^-$-induced cAMP increase does not affect $[Ca^{2+}]_i$ in mouse (*Wennemuth et al., 2003*; *Carlson et al., 2007*) and human sperm (*Strünker et al., 2011*). In contrast, 8-Br-cAMP, 8-Br-cGMP, and db-cAMP evoked a $Ca^{2+}$ signal in wild-type and bPAC sperm (*Figure 3A–C,E*) that was abolished in *CatSper*-null sperm (*Figure 3F*); similar results have been reported by others (*Ren et al., 2001*; *Xia et al., 2007*). In human sperm, 8-Br-cNMPs directly activate CatSper via an extracellular site (*Brenker et al., 2012*). Experiments

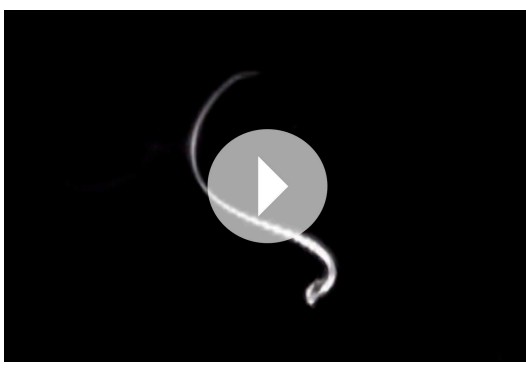

**Video 1**. Wild-type sperm before and after UV flash. The cell was tethered to the glass surface by lowering the BSA concentration (0.3 mg/ml). The recording was performed using an epifluorescent microscope (IX71; Olympus) equipped with a dark-field condenser and a 10x objective (UPlanFL, NA 0.3; Olympus) and an additional 1.6× lenses. Frames were acquired at 200 fps using a CMOS camera (Dimax; PCO). Stimulation with UV light for 200 ms was achieved using a UV LED. For clarity, the video is played with 100 fps (half original speed).

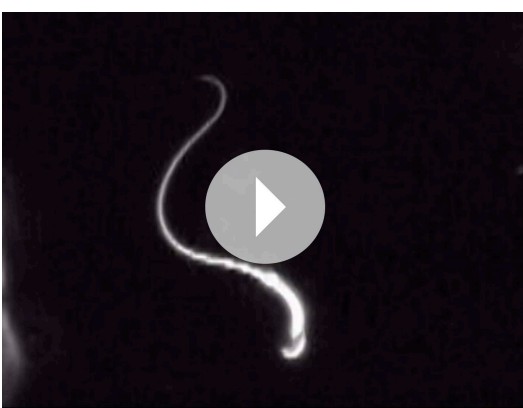

**Video 2**. bPAC sperm before and after UV flash. The recording was performed as described for **Video 1**.

analyzing the action of 8-Br-cNMPs on membrane currents in mouse sperm yielded inconsistent results (*Kirichok et al., 2006*; *Cisneros-Mejorado et al., 2014*). Thus, this effect needs to be addressed by future studies.

## Light-stimulated cAMP synthesis restores fertility

The loss of SACY function abolishes cAMP signaling in sperm, rendering sperm immotile and resulting in male infertility (*Esposito et al., 2004*; *Xie et al., 2006*). We explored the potential of optogenetics to compensate for the functional loss of SACY. Removal of extracellular $Ca^{2+}$ attenuated $HCO_3^-$-induced cAMP synthesis by SACY (*Figure 4A*) (*Jaiswal and Conti, 2003*; *Carlson et al., 2007*). Thereby, tyrosine phosphorylation in wild-type and bPAC sperm was largely abolished (*Figure 4B*). Elevation of basal cAMP levels by the phosphodiesterase inhibitor IBMX restored tyrosine phosphorylation in wild-type sperm (*Figure 4A,B*). Similarly, in $Ca^{2+}$-free medium, the light-stimulated increase of cAMP levels restored tyrosine phosphorylation in bPAC sperm (*Figure 4A,B*), demonstrating that bPAC activation compensates for the loss of SACY activity. Furthermore, we examined whether in sperm lacking functional SACY, bPAC activation can restore motility and the ability to fertilize the egg. To this end, we crossed Prm1-bPAC mice with mice lacking the sperm-specific $Na^+/H^+$ exchanger sNHE, encoded by the *Slc9a10* gene. Male *Slc9a10*-null mice are infertile, because their sperm are largely immotile (*Wang et al., 2003*). In *Slc9a10*-null mice, the $HCO_3^-$-stimulated SACY activity is abolished as well, probably because sNHE and SACY form a functional signaling complex that is disrupted in these mice (*Wang et al., 2007*). The motility of *Slc9a10*-null sperm is restored by db-cAMP (*Wang et al., 2003*); however, db-cAMP not only elevates cAMP levels but also evokes a $Ca^{2+}$ influx via CatSper (*Figure 3E*). Thus, we tested whether motility can be restored by solely increasing cAMP levels with light. Sperm were immotile in darkness (*Figure 4C*, left); light stimulation restored both the flagellar beat and forward motility (*Figure 4C*, right; *Video 5*), demonstrating that restoring cAMP levels alone is sufficient to rescue motility. We went one step further and tested whether *Slc9a10*-null/bPAC sperm also regain their fertilization potential upon light stimulation. Indeed, light-stimulated *Slc9a10*-null/bPAC sperm were able to fertilize oocytes in vitro (*Figure 4D*), demonstrating that optogenetics can restore fertility.

## Discussion

In this study, we report on a transgenic mouse model designed to control cAMP signaling in sperm using optogenetics. We demonstrate that transgenic expression of bPAC in sperm does not interfere with the endogenous $HCO_3^-$-stimulated SACY activity. In fact, bPAC mimics the effect of SACY activity: the light-stimulated increase in cAMP levels is similar to that activated by $HCO_3^-$ and activation of bPAC increases the flagellar beat frequency in a dose-dependent manner. Thus, the Prm1-bPAC mouse

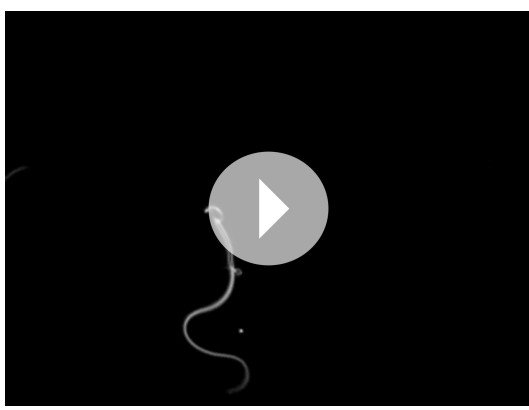

**Video 3**. Local illumination of bPAC sperm. The recording was performed as described for **Video 1** using local illumination of the sperm flagellum for 100 ms; the video is played with 100 fps (half of the original speed). In this setting, the light flash is not visible and is, therefore, visualized with a circle. The analysis of the beat frequency is presented in **Figure 2K**.

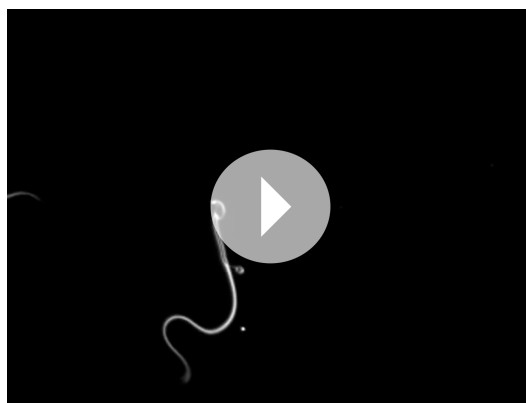

**Video 4**. Local illumination of bPAC sperm. **Video 3** shown at higher intensity for the frames containing the light flash. This allows to visualize the light flash. The profile of the light flash is indicated in an inset. For clarity, the video speed was reduced to 10% of the frames containing the light flash.

model is an ideal optogenetic tool to study cAMP signaling in sperm.

Furthermore, we show that optogenetics can be used to restore fertility. Sperm from infertile *Slc9a10*-null mice are devoid of cAMP signaling, because they lack $HCO_3^-$-stimulated SACY activity, which is essential for sperm motility (**Wang et al., 2003**, **2007**). Expression of bPAC in *Slc9a10*-null sperm allowed restoring motility after light stimulation and, most importantly, restoring fertilization in vitro. Of note, in vitro fertilization (IVF) using light-stimulated sperm was successful using zona pellucida-intact oocytes, demonstrating that a cAMP increase, at least in vitro, is sufficient to restore fertility in *Slc9a10*-null mice. In contrast, incubation of *Slc9a10*-null sperm with cAMP analogs only restored fertilization of zona pellucida-free but not zona pellucida-intact oocytes (**Wang et al., 2003**). This discrepancy could be explained by (1) cAMP analogs not reaching sufficiently high levels in sperm, or (2) cAMP analogs being not as efficient as cAMP to stimulate downstream targets, or (3) non-specific effects compromising the interpretation of their action (**Brenker et al., 2012**). The Prm1-bPAC mouse model holds great promise studying cAMP signaling uncompromised by side effects and allowed us to solve a long-standing controversy by showing that cAMP, in fact, does not stimulate a $Ca^{2+}$ influx into mouse sperm (**Figure 3B**). These results demonstrate that optogenetics can disentangle cellular events such as $Ca^{2+}$ and cAMP signaling in vivo.

Now that we have established the Prm1-bPAC mouse model and demonstrated that it can be used to analyze cAMP signaling in sperm, the next step will be to implement the spatio-temporal precision of this tool to study cAMP signaling in sperm. For example, $HCO_3^-$ stimulation evokes a rapid cAMP increase and changes the flagellar beat frequency within seconds. However, PKA-dependent activation of tyrosine phosphorylation is barely detectable within the first hour of $HCO_3^-$ incubation (**Visconti et al., 1995a**). Thus, $HCO_3^-$ exhibits short-term and long-term effects and probably controls cellular events in addition to cAMP synthesis via SACY. The Prm1-bPAC mouse model allows deciphering these signaling events, if bPAC expression does not interfere with other signaling pathways.

Signaling molecules in sperm flagella are highly compartmentalized (**Chung et al., 2014**). The flagellar beat is controlled by cAMP-dependent phosphorylation of PKA targets in the axoneme (**Salathe, 2007**); however, it is unknown whether cAMP signaling is restricted to microdomains or whether changes in cAMP propagate along the flagellum. Compartmentalization of cAMP signaling confers specificity to this ubiquitous cellular messenger, for example, in cell-cycle progression or modulation of gene expression. (**Michel and Scott, 2002**; **Zaccolo and Pozzan, 2002**; **Willoughby and Cooper, 2007**). Our results show that local illumination of the flagellum of bPAC sperm is sufficient to increase its beat frequency. Optogenetics using bPAC provides the spatial and temporal precision to increase cAMP levels in subcellular regions and specialized signaling compartments such as cilia and flagella,

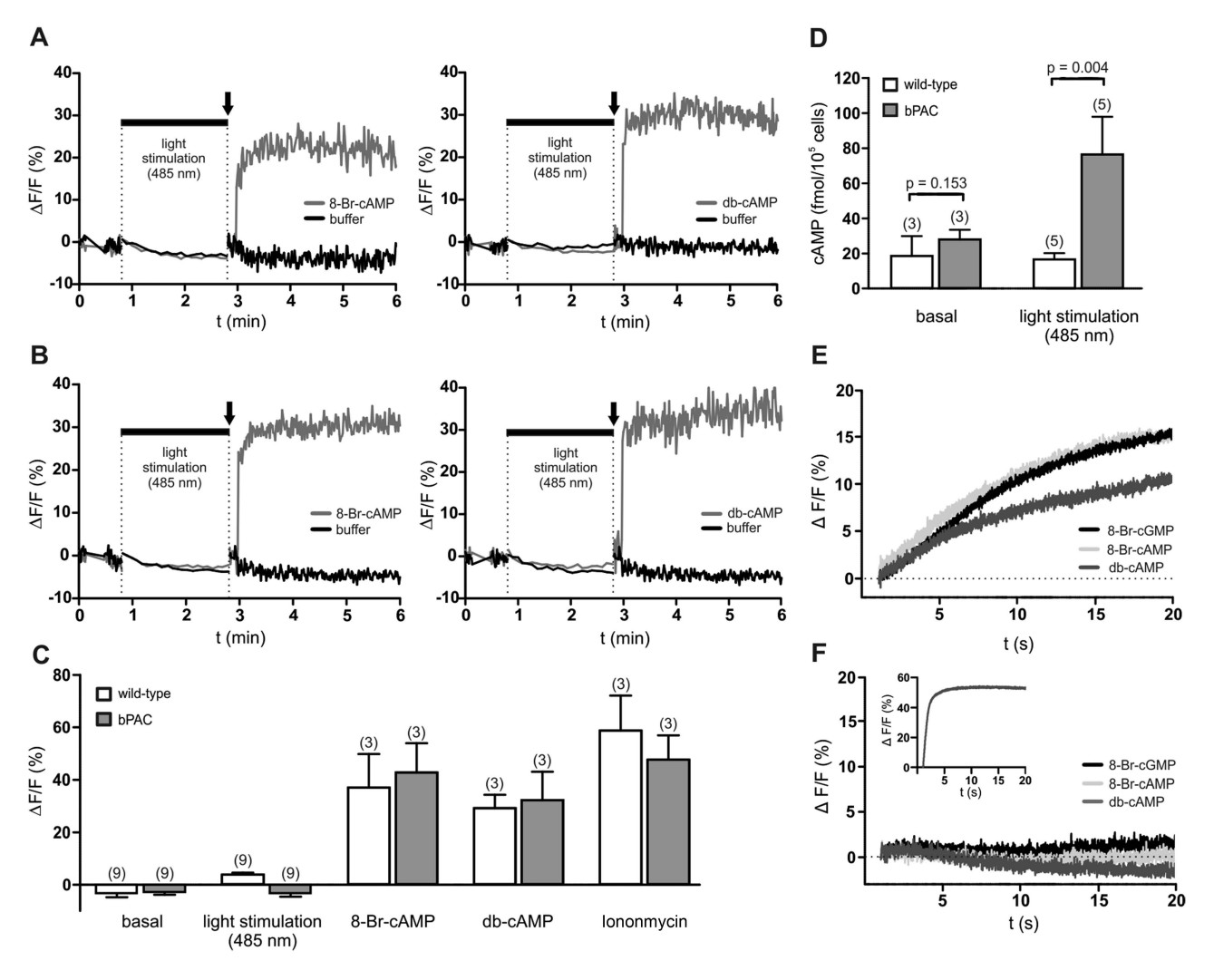

**Figure 3**. cAMP does not evoke a $Ca^{2+}$ influx in mouse sperm. (**A**, **B**) $Ca^{2+}$ signals induced by 8-Br-cAMP (10 mM; left) and db-cAMP (10 mM; right) in wild-type (**A**) and bPAC sperm (**B**) loaded with the fluorescent $Ca^{2+}$ indicator FluoForte. Arrows indicate the addition of compounds. Light stimulation does not evoke a $Ca^{2+}$ signal. Signals were measured in 96 multi-well plates in a fluorescence plate reader. (**C**) Mean signals evoked by light stimulation, cyclic nucleotide analogs, and ionomycin. (**D**) Illumination according to the protocol in (**A**, **B**) stimulates cAMP synthesis in bPAC sperm. Data are plotted as mean ± s.d.; (n) = number of experiments; p values calculated using Student's t test. (**E**, **F**) $Ca^{2+}$ signals of capacitated sperm induced by cyclic nucleotide analogs (10 mM) in wild-type (**E**) and *CatSper*-null (**F**) sperm loaded with the fluorescent $Ca^{2+}$ indicator Cal-520 in a stopped-flow device. Inset in (**F**): ionomycin control (2 µM).

which allows to study complex signaling networks - not only in mammalian sperm but also any other ciliated cell type.

Finally, the optogenetic toolkit has been recently expanded by a light-activated phosphodiesterase (LAPD) that hydrolyses cyclic nucleotides upon stimulation by red light (*Gasser et al., 2014*). A combination of LAPD and bPAC with fluorescent cAMP biosensors holds great promise to map the dynamics of cAMP signaling in live cells in precise spatio-temporal and quantitative terms. These findings encourage future studies to explore the full potential of controlling cellular messengers by optogenetics.

## Materials and methods

### Generation and genotyping of Prm1-bPAC transgenic mice

The bPAC cDNA (beta subunit of photo-activated adenylyl cyclase from *Beggiatoa* sp. PS, BGP_1043; modified for expression in mammalian systems according to *Stierl et al. (2011)*, GU461307) sequence

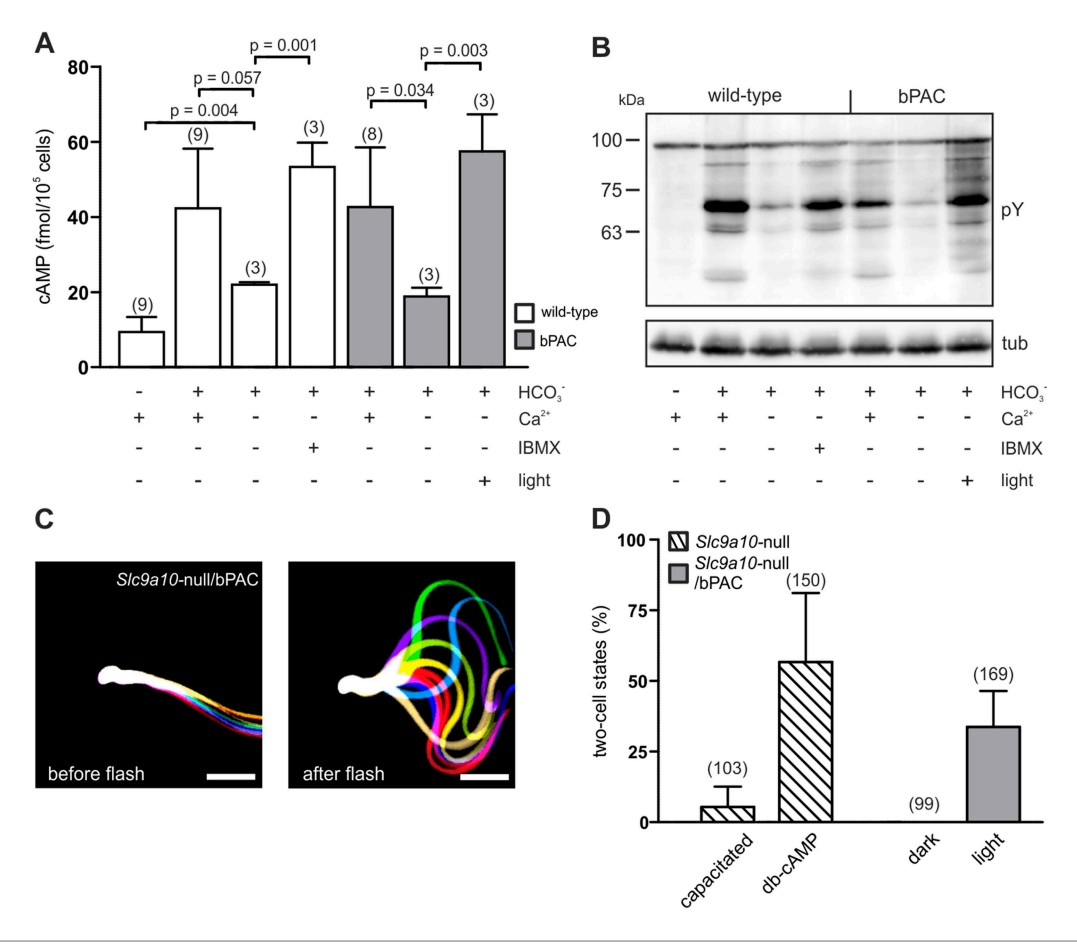

**Figure 4.** bPAC restores fertility in mice lacking functional SACY. (**A**) Nominally $Ca^{2+}$-free medium attenuates $HCO_3^-$-induced cAMP synthesis by SACY in wild-type and bPAC sperm. Inhibition of phosphodiesterases by IBMX in wild-type sperm and light stimulation in bPAC sperm increases cAMP levels without extracellular $Ca^{2+}$. Data are plotted as mean ± s.d.; (n) = number of experiments; p values calculated using Student's t test. (**B**) In the absence of extracellular $Ca^{2+}$, $HCO_3^-$-induced tyrosine phosphorylation (pY) is strongly attenuated. In bPAC sperm, light stimulation is sufficient to restore tyrosine phosphorylation. (**C**) Light stimulation restores flagellar beating of *Slc9a10*-null/bPAC sperm. Flagellar waveform of *Slc9a10*-null/bPAC sperm before (left) and after light stimulation (right). Successive, aligned, and superimposed images creating a 'stop-motion' image, illustrating one flagellar beating cycle. Scale bar: 30 µm. (**D**) Upon light stimulation, *Slc9a10*-null/bPAC sperm fertilize oocytes in vitro (mean ± s.d.; (n) = total number of oocytes from three independent experiments).

was amplified via PCR. A hemagglutinin (HA) tag was fused to the C terminus, an *EcoRI* restriction site was added to the 5′ end and a *BamHI* and *XhoI* restriction site to the 3′ end by nested PCR. The PCR product was cloned into a pBluescript SK(-) vector (Agilent Technologies, Santa Clara, USA) using *EcoRI* and *XhoI* (pB-bPAC). After sequencing, the bPAC-HA insert was excised and cloned into pPrCExV (a kind gift from Robert Braun, Jackson Laboratory) using *EcoRI/BamHI*. Transgenic mice were generated via pronuclear injection following standard procedures (***Ittner and Götz, 2007***) at the HET animal facility (University Hospital Bonn, Germany). Mice were genotyped by PCR using bPAC-specific primers (see also ***Figure 1A***). Mice used in this study were 2–5 months of age. All animal experiments were in accordance with the relevant guidelines and regulations.

### *Slc9a10*-null mice

*Slc9a10*-null mice were purchased from the Jackson Laboratory (B6;129S6-*Slc9a10tm1Gar*/J, stock number: 007661).

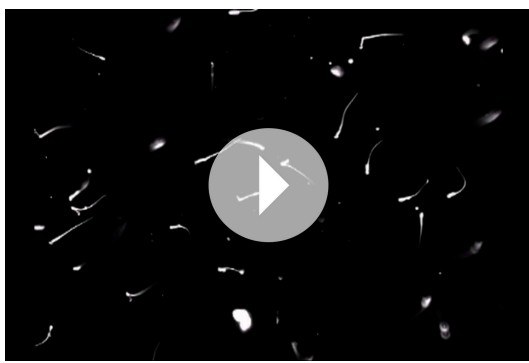

**Video 5**. Slc9a10-null/bPAC sperm before and after UV flash. The recording was performed as described for *Video 1*, but without the additional 1.6× magnification, an acquisition frequency of 90 fps, and 500 ms UV stimulation. The video is shown in real time.

## Western blot analysis

For heterologous expression, HEK293 cells were transfected using Lipofectamine 2000 (Life Technologies, Carlsbad, USA). Lysates were obtained by homogenizing cells or tissues in lysis buffer (10 mM Tris/HCl, pH 7.6, 140 mM NaCl, 1 mM EDTA, 1% Triton X-100, mPIC protease inhibitor cocktail 1:500) followed by trituration through a 18-gauge needle. Samples were incubated for 30 min on ice and centrifuged at $10,000 \times g$ for 5 min at 4°C. Prior to separation by SDS-PAGE, samples were mixed with 4× SDS loading buffer (200 mM Tric/HCl, pH 6.8, 8% SDS (wt/vol), 4% β-mercaptoethanol (vol/vol), 50% glycerol, 0.04% bromophenol blue) and heated for 5 min at 95°C. To activate bPAC, sperm samples were illuminated for 90 min in suspension in a reaction tube using a halogen fiber optics illuminator (Olympus, Tokio, Japan). Sperm samples used for SDS-PAGE were washed with 1 ml PBS and sedimented by centrifugation at $5000 \times g$ for 5 min. $1–2 \times 10^6$ cells were resuspended in 4× SDS loading buffer and heated for 5 min at 95°C. For Western blot analysis, proteins were transferred onto PVDF membranes (Merck Millipore, Billerica, USA), probed with antibodies, and analysed using a chemiluminescence detection system. For re-probing, membranes were incubated in stripping buffer (62.5 mM Tris/HCl pH 6.7, 2% SDS) for 30 min at 65°C and washed with PBS before incubation with a new antibody. For the study of protein phosphorylation, sperm were incubated in capacitating buffer containing 25 mM NaHCO₃ for 10 min (PKA-dependent phosphorylation) or 90 min (tyrosine kinase-dependent phosphorylation). Similarly, bPAC sperm were illuminated for either 10 or 90 min using a halogen fibre optics illuminator (150 W, Intralux 6000-1; Volpi, Schlieren, Switzerland).

Primary antibodies: anti-HA 3F10 (1:5000; Roche, Basel, Switzerland), anti-SACY R21 (1:2000; CEP Biotech, Tamarac, USA), anti-phosphotyrosine 4G10 (1:1000; Merck Millipore), anti-PKA[C] 5B (1:4000; BD Transduction Laboratories, San Jose, USA), anti-phospho-PKA substrate (1:1000; Cell Signaling, Danvers, USA), anti-α-tubulin (1:5000; Sigma-Aldrich, Seelze, Germany); secondary antibodies: goat-anti-rat, HRP conjugated (1:5000, Dianova, Hamburg, Germany), sheep-anti-mouse, HRP conjugated (1:5000, Dianova).

## Immunohistochemistry

Testes were fixed in 4% paraformaldehyde/PBS overnight, cryo-protected in 10 and 30% sucrose, and embedded in Tissue-Tek (Sakura Finetek, Alphen aan den Rijn, Netherlands). To block unspecific binding sites, cryosections (16 µm) were incubated for 1 hr with blocking buffer (0.5% Triton X-100 and 5% ChemiBLOCKER (Merck Millipore) in 0.1 M phosphate buffer, pH 7.4). The primary anti-HA antibody (rat monoclonal; Roche) was diluted 1:1000 in blocking buffer and incubated for 2 hr. Fluorescent secondary antibodies (donkey anti-rat Alexa488; Dianova) was diluted 1:400 in blocking buffer containing 0.5 mg/ml DAPI (Life Technologies) and pictures were taken on a confocal microscope (FV1000; Olympus).

## Sperm preparation

Sperm were isolated by incision of the cauda followed by a swim-out in modified TYH medium (135 mM NaCl, 4.8 mM KCl, 2 mM CaCl₂, 1.2 mM KH₂PO₄, 1 mM MgSO₄, 5.6 mM glucose, 0.5 mM sodium pyruvate, 10 mM L-lactate, 10 mM HEPES, pH 7.4). For capacitation, the medium contained 3 mg/ml BSA and 25 mM of the NaCl was substituted with 25 mM NaHCO₃. The pH was adjusted at 37°C. After 15–30 min swim-out at 37°C, sperm were collected and counted. Red light was used during the isolation of bPAC sperm.

## Determination of cAMP content

Sperm were adjusted to a concentration of $1.25 \times 10^7$ cells per ml with TYH buffer containing different compounds with the following final concentrations: 25 mM NaHCO₃, 1 mM N⁶,2′-O-dibutyryladenosine-3′,5′-cyclic monophosphate (db-cAMP; Sigma-Aldrich), 0.5 mM IBMX (Sigma-Aldrich). To activate

bPAC, sperm samples were illuminated in suspension in a reaction tube using a halogen fiber-optics illuminator (150 W, Intralux 6000-1; Volpi, Schlieren, Switzerland). After stimulation with light or compounds for 1 to 10 min, the reaction was quenched with $HClO_4$ (1:3 (vol/vol); 0.5 M final concentration). Samples were frozen in liquid $N_2$, thawed, and neutralized by addition of $K_3PO_4$ (0.24 M final concentration). The salt precipitate and cell debris were sedimented by centrifugation (15 min, 15,000 g, 4°C). The cAMP content in the supernatant was determined by a competitive immunoassay (Molecular Devices, Sunnyvale, USA), including an acetylation step for higher sensitivity. Calibration curves were obtained by serial dilutions of cAMP standards. 96-well plates were analysed by using a microplate reader (FLUOstar Omega; BMGLabtech, Ortenberg, Germany).

## Analysis of acrosomal exocytosis

$1 \times 10^6$ sperm were capacitated in TYH buffer supplemented with 3 mg/ml BSA and 25 mM $NaHCO_3$. Non-capacitated sperm were incubated in BSA buffer only. To induce capacitation by light, sperm were illuminated with a halogen fiber-optics illuminator (150 W, Intralux 6000-1; Volpi) at 37°C for 90 min. The acrosome reaction was induced by incubating sperm with 2 µM ionomycin for 10 min, incubation with 1% DMSO served as control. Sperm were collected by centrifugation and resuspended in 100 µl PBS buffer. Samples were air dried on microscope slides and fixed with 100% ethanol for 30 min at room temperature. For acrosome staining, sperm were labeled with 5 µg/ml PNA-FITC (L7381, Sigma-Aldrich) and 2 µg/ml DAPI in PBS for 30 min. Images were acquired using a confocal laser scanning microscope (FV1000; Olympus), a minimum of 600 cells was analyzed per condition.

## Sperm motility analysis

Sperm motility was studied in shallow observation chambers (depth 150 µm). Cells were tethered to the glass surface by adjusting the BSA concentration in the buffer to 0.3 mg/ml. For analysis, cells that had their head attached to the glass surface and that had a free beating flagellum were chosen. Sperm motility was recorded under an inverted microscope (IX71; Olympus) equipped with a dark-field condenser, a 10x objective (UPlanFL, NA 0.3; Olympus), and additional 1.6× lenses (16× final magnification). The temperature of the microscope incubator (Life Imaging Services, Basel, Switzerland) was adjusted to 37°C. To obtain sharp images of moving sperm, stroboscopic illumination (2 ms light pulses) was achieved using a red LED (M660L3-C1; Thorlabs, Newton, USA) with a custom-made power supply. The camera and the LED light pulses were synchronized using a function generator (33220A; Agilent). Images were collected at 200 frames per second using a CMOS camera with a pixel size of 11 µm (Dimax; PCO, Kelheim, Germany). Stimulation of bPAC was achieved using a collimated UV LED (365 nm; ~12 mW) coupled to the epifluorescence port of the microscope (M365L2-C1; Thorlabs) and a TTL-controlled custom-made power supply. Irradiation time was set using a TTL pulse generator (UTG100; ELV). For local activation of the flagellum of bPAC sperm, we partially focused the UV-LED onto a 100-µm diaphragm (Linos Photonik) by detuning its corresponding collimator lens. The diaphragm aperture was then imaged onto the objective focal plane by a plano-convex lens (f = 100 mm; LA1509A, Thorlabs). This resulted in a Gaussian-shaped UV spot ($R^2$ of the fit 0.96) with a 12-µm width at the focal plane. The total UV-light power delivered was 2.3 mW. Quantification of the flagellar beat was performed using custom-made programs written in MATLAB (MathWorks. Natick, USA). The software can be made available upon request. The program identified the best threshold for binarization followed by a skeleton operation to identify the flagellum. The flagellar beat parameters were determined within a time window of 0.5 s before and after each frame. For frames at the boundaries (the beginning or the end of the video or flanking the UV flash), the time windows were asymmetric but contained the same number of frames. We monitored the angle between the straight line connecting the middle of the flagellum with the sperm head and the axis of symmetry of the cell. This angle varied in a sinusoid-like manner in time. The beat frequency was obtained by fitting a sinus to this wave. For alignment of the flagellar beat envelopes, we used custom-made programs written in LabVIEW. Using defined thresholds, the image was binarized. From a user-defined region-of-interest centred at the cell head, the program determined the location of the head on subsequent frames using a registering procedure. The neck of the cell was identified by applying a mask with the shape of an annulus centred into the sperm head. The annulus had an internal diameter of 16 µm to cover the sperm head and a 4 µm longer external diameter, enough to resolve the first pixels of the neck. All frames were then rotated and superimposed with a rotation angle equal to the azimuth of the neck region on a reference system centred at the sperm head. For better visualization, representative

datasets were smoothed using Graph Pad Prism 5.02 (factor 20 for global illumination, factor 100 local illumination).

## Measurement of changes in intracellular $Ca^{2+}$

Changes in $[Ca^{2+}]_i$ were recorded using the fluorescent $Ca^{2+}$ indicators Cal-520, AM (AAT Bioquest, Sunnyvale, USA) and FluoForte (ENZO Life Sciences, Lörrach, Germany) in 96 multi-well plates in a fluorescence plate reader (Fluostar Omega; BMGLabtech) and in a rapid-mixing device in the stopped-flow mode (SFM400; Bio-Logic, Claix, France). For $Ca^{2+}$ measurements in the plate reader, sperm were loaded with FluoForte (20 µM) in the presence of Pluronic F-127 (0.02% vol/vol) at 37°C for 45 min. After incubation, excess dye was removed by three centrifugation steps (700×$g$, 7 min, room temperature). The pellet was resuspended in TYH and equilibrated for 5 min at 37°C. Each well was filled with 100 µl ($10^6$ sperm ml$^{-1}$) of the sperm suspension. To record changes in $[Ca^{2+}]_i$ upon activation of bPAC, samples were alternately excited at 485 nm and 520 nm. As a control, fluorescence was recorded before and after the addition of 10 µl of different compounds to the final concentration of 10 mM (db-cAMP, 8-Bromo-cAMP, 8-Bromo-cGMP; Sigma-Aldrich). Changes in fluorescence are depicted as ΔF/F0 (%), indicating the percentage change in fluorescence (ΔF) with respect to the mean basal fluorescence (F0) before application of buffer or compounds.

To record changes in $[Ca^{2+}]_i$ in a rapid-mixing device (SFM-400; Bio-Logic) in the stopped-flow mode, sperm were loaded with Cal-520 (5 µM) as described above for FluoForte. Changes in $[Ca^{2+}]_i$ were measured as previously described (*Strünker et al., 2011*) with minor modifications. In brief, the sperm suspension ($5 \times 10^6$ sperm/ml) was rapidly mixed 1:1 (vol/vol) with the respective stimulants at a flow rate of 0.5 ml/s. Fluorescence was excited by a SpectraX Light Engine (Lumencor, Beaverton, USA), whose intensity was modulated with a frequency of 10 kHz. The excitation light was passed through a BrightLine 475/28 nm filter (Semrock, Rochester, USA) onto the cuvette. Emission light was passed through a BrightLine 536/40 filter (Semrock) and recorded by photomultiplier modules (H10723-20; Hamamatsu Photonics). The signal was amplified and filtered through a lock-in amplifier (7230 DualPhase; Ametek, Paoli, USA). Data acquisition was performed with a data acquisition pad (PCI-6221; National Instruments. Austin, USA) and Bio-Kine software v. 4.49 (Bio-Logic). $Ca^{2+}$ signals are depicted as the percent change in fluorescence (ΔF) with respect to the mean of the first three data points recorded immediately after mixing (F0), that is, when a stable fluorescence signal was observed. The control (buffer) ΔF/F0 signal was subtracted from compound-induced signals.

## In vitro fertilization

Superovulation in females was induced by intraperitoneal injection of 10 I.U. Intergonan (SimposiumVet, Lisbon, Portugal) 3 days before the experiment. 14 hours before oocyte preparation, mice were injected with 10 I.U. Ovogest (SimposiumVet). KSOM medium (EmbryoMax Modified M16 Medium; Merck Millipore) was mixed 1:1 with mineral oil (Sigma-Aldrich) and equilibrated overnight at 37°C. On the day of preparation, 100 µl drops of KSOM medium were covered with the medium/oil mixture and 100,000 sperm were added to each drop. Sperm were capacitated for 90 min in TYH medium supplemented as indicated above. For stimulation of bPAC, the medium contained no $HCO_3^-$ and reaction tubes containing sperm were placed in a custom-made rack equipped with blue LEDs for 90 min. Cumulus-enclosed oocytes were prepared from the oviducts of superovulated females and added to the drops. After 4 hr at 37°C and 5% $CO_2$, oocytes were transferred to fresh KSOM medium. The number of 2-cell stages was evaluated after 24 hr.

## Acknowledgements

We thank N Blank, JH Krause, R Pascal, and I Lux for technical assistance, W Bönigk for cloning, and R Braun (Jackson Laboratory) for providing the pPrCExV vector.

## Additional information

### Funding

| Funder | Grant reference number | Author |
| --- | --- | --- |
| Max-Planck-Gesellschaft | Minerva Program | Dagmar Wachten |

| Funder | Grant reference number | Author |
| --- | --- | --- |
| University of Bonn | Maria-von-Linden Program | Dagmar Wachten |
| Deutsche Forschungsgemeinschaft | SFB645 (INST 217/555-2) | Timo Strünker, U Benjamin Kaupp, Dagmar Wachten |
| Deutsche Forschungsgemeinschaft | Bonn Excellence Cluster ImmunoSensation | Luis Alvarez, U Benjamin Kaupp, Dagmar Wachten |

The funders had no role in study design, data collection and interpretation, or the decision to submit the work for publication.

## Author contributions

VJ, Conception and design, Acquisition of data, Analysis and interpretation of data, Drafting or revising the article; LA, Conception and design, Analysis and interpretation of data, Contributed unpublished essential data or reagents; MB, Acquisition of data, Analysis and interpretation of data; TS, DW, Conception and design, Analysis and interpretation of data, Drafting or revising the article; PH, Drafting or revising the article, Contributed unpublished essential data or reagents; UBK, Conception and design, Drafting or revising the article

## Ethics

Animal experimentation: This study was performed in strict accordance with the recommendations in the Guide for the Care and Use of Laboratory Animals of the LANUV (Landesamt für Natur, Umwelt und Verbraucherschutz). Reference number: 84-02.04.2012.A192.

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
