## [Decision Letter]

Thank you for sending your work entitled “Controlling fertilization and cAMP signaling in sperm flagella by optogenetics” for consideration at *eLife*. Your article has been favorably evaluated by a Senior editor, David Clapham as a guest Reviewing editor, and two peer reviewers.

The Reviewing editor and the reviewers discussed their comments before we reached this decision, and the Reviewing editor has assembled the following comments to help you prepare a revised submission.

The full reviewers comments are shown at the end of this letter, but the points for revision boil down to the following:

1) You need to more convincingly show that light stimulation in the imaging experiments induces a significant [cAMP] increase in the sperm principal piece.

2) You need to show that cAMP/cGMP analogs activate ICatSper in mouse sperm. You state: “However, these analogues failed to induce a Ca^2+^ influx in CatSper-null sperm (Figure 3) (28), indicating that analogues of cyclic nucleotides, like in human sperm, nonspecifically activate also mouse CatSper. These results solve a long-standing controversy concerning the interconnection of cAMP and Ca^2+^ influx in mammalian sperm.” You assume that 8-Br-cAMP/cGMP also activates ICatSper in mouse sperm, “like in human sperm”. Although not referenced in this sentence, the authors are presumably referring to their previous paper (EMBOJ, 2012, 31,1654-1665, Figure 8) where high concentration (5 mM) 8-Br-cGMP enhanced a monovalent ICatSper-like current by ∼ 2 fold. Such a small change at this high dose is difficult to interpret. In the Kirichok et al. paper (Nature, 2006, 439: 737-740), 8-Br-cAMP was tested on mouse sperm, and there was no ICatSper activation. Thus, if you wish to support this statement, you could record mouse ICatSper changes with 8Br-cAMP and show that 8-Br-cAMP activation is nonspecific, perhaps by showing a dose response curve, comparing inside and outside perfusion of the analog, etc.

Alternatively, for both points, you could refrain from overextending your conclusions and simply state the points you have definitively shown. This would require removing the comments quoted above and alike comments in the discussion, and state without exaggeration the merits and current limitations of the experiments. This is an important new tool, and it is best to accurately state what is, and is not, possible to conclude. Further refinement of the indicator, or other indicators may enable you to definitively address these points in the future.

*Reviewer #1*:

In this manuscript, Jansen et al. made a transgenic mouse line expressing a bacterial photoactivated adenylyl cyclase (bPAC) in testis and sperm. bPAC was previously developed and tested by the Hegemann group in *Xenopus oocytes* and cultured neurons; this paper reports the first use of bPAC in whole animals. Light stimulation in the transgenic sperm is able to increase cAMP concentration, leads to the acrosome reaction and triggers sperm tail beating. In addition, light stimulation also rescues the fertililization deficiency in the sNHE (a sperm-specific putative Na^+^/H^+^ exchanger) knockout sperm.

The tool developed in the paper allowed the authors to directly test the function of cAMP. Previous experiments relied on the addition of bicarbonate, exogenously applied cell-permeable cAMP or cAMP UV-uncaging, which can be somewhat indirect. The results from the paper largely confirmed previous findings: cAMP is important for all the major sperm physiology including capacitation, motility and fertilization. One apparently surprising finding is that light stimulation does not lead to obvious increase of intracellular Ca^2+^ concentration.

The experiments in the paper are well designed and the data is convincing. This is the first reported optogenetic tool applied to sperm physiology. The tool will allow cAMP signaling studies with high temporal and spatial resolution in the future.

The only major comment I have is on the conclusion that cAMP does not induce Ca^2+^ influx ([Ca^2+^]i increase) in sperm, which is apparently at odd with various previous studies with “indirect” methods. While the finding in Figure 3, that light stimulation does not lead to detectable change in Δ F/F, is consistent with the idea that cAMP doesn't cause Ca^2+^ influx, there can be other interpretations to the data. For example, the light-induced [cAMP] increase is relatively moderate (∼2 to 4 fold) and is comparable to that induced by HCO_3_ (Figure 2). Earlier studies by Babcock and colleagues (e.g. Wennemuth, JBC, 2000) showed that large detectable HCO_3_^-^ induced [Ca^2+^] increases were obvious only when combined with K8.6 stimulus (higher [K+] with alkali bath). It is possible that at the level achieved by light stimulation in the current study, cAMP requires other co-stimuli to induce Ca^2+^ influx. Another possibility, perhaps even more likely, is that the 2-4 fold [cAMP] increase, which is measured from whole, is largely from sperm head (because of its overwhelmingly larger volume) but little from sperm principal piece where the Ca^2+^ influx channels are localized. The experiments in the paper do not show whether there is [cAMP] increase in sperm tail upon light stimulation. The geometry of the sperm midpiece and the filled mitochondria perhaps are significant barriers for cAMP diffusion from sperm head to tail.

*Reviewer #2*:

This paper describes a novel animal model featuring expression of the light-activated adenylyl cyclase bPAC exclusively in sperm. The authors present exciting data showing rapid changes in flagellar beat in live sperm upon stimulation of bPAC, and that light can serve as a surrogate for other stimuli believed to elevate cAMP in sperm such as HCO_3_^-^ treatment. The changes in flagellar beat also provide a useful (albeit indirect) readout of the temporal features of illumination-dependent cAMP production by bPAC.

This optogenetic tool will provide a fantastic opportunity to clarify the role of compartmentalized cAMP signaling in sperm motility, maturation and fertilization. The animal model is well-validated (Figure 1), and bPAC expression does not alter expression of other cAMP signaling proteins and pathways (Figure 2). While expression of bPAC did not alter PKA phosphorylation per se (Figure 2), it would have been nice to see that illumination could also produce sufficient cAMP to stimulate phosphorylation of PKA targets. If the authors could produce such a figure, it would strengthen the paper.

In Figure 3, the authors confirm previous findings that cyclic nucleotide analogs (10 mM 8Br-cAMP and 10mM db-cAMP) caused Ca^2+^ entry into the sperm via CatSper, but light stimulation of bPAC did not cause this Ca^2+^ entry. Is another possible interpretation of this result simply that the illumination of bPAC didn't cause sufficient intracellular cAMP production to influence CatSper? On the other hand, the concentration of cAMP analogs in the present study is much higher than would typically be used in most other cell types to activate cAMP targets. Is this concentration comparable to what was used previously in other studies that examined the effects of cAMP analogs on calcium entry? What was the concentration of db-cAMP used in Figure 2 (cAMP-dependent tyrosine phosphorylation)?

---

## [Author Response]

*1) You need to more convincingly show that light stimulation in the imaging experiments induces a significant [cAMP] increase in the sperm principal piece*.

We would love to do this experiment. Such an experiment requires the transgenic expression of a cAMP FRET biosensor in the flagellum of intact sperm; the spectral properties of the cAMP sensor and bPAC must be orthogonal. Such a sensor is not yet available. Notwithstanding, we show in a new data set (Figure 2 and Videos 4 and 5) that stimulation of a small part of the flagellum with a single flash increased the flagellar beat frequency, demonstrating that local bPAC activation significantly increases cAMP levels in the flagellum. We have amended the Results, Discussion, and Materials and methods sections accordingly and added a sentence to the discussion stating that it would be ideal to combine optogenetic tools with a fluorescent cAMP biosensor to study cAMP dynamics in live cells.

*2) You need to show that cAMP/cGMP analogs activate ICatSper in mouse sperm. You state: “However, these analogues failed to induce a Ca*^*2+*^
*influx in CatSper-null sperm (*Figure 3*) (*[28]*), indicating that analogues of cyclic nucleotides, like in human sperm, nonspecifically activate also mouse CatSper. These results solve a long-standing controversy concerning the interconnection of cAMP and Ca*^*2+*^
*influx in mammalian sperm.” You assume that 8-Br-cAMP/cGMP also activates ICatSper in mouse sperm, “like in human sperm”. Although not referenced in this sentence, the authors are presumably referring to their previous paper (EMBOJ, 2012, 31,1654-1665, Figure 8) where high concentration (5 mM) 8-Br-cGMP enhanced a monovalent ICatSper-like current by ∼ 2 fold. Such a small change at this high dose is difficult to interpret. In the Kirichok et al. paper (Nature, 2006, 439: 737-740), 8-Br-cAMP was tested on mouse sperm, and there was no ICatSper activation. Thus, if you wish to support this statement, you could record mouse ICatSper changes with 8Br-cAMP and show that 8-Br-cAMP activation is nonspecific, perhaps by showing a dose response curve, comparing inside and outside perfusion of the analog, etc*.

*Alternatively, for both points, you could refrain from overextending your conclusions and simply state the points you have definitively shown. This would require removing the comments quoted above and alike comments in the discussion, and state without exaggeration the merits and current limitations of the experiments. This is an important new tool, and it is best to accurately state what is, and is not, possible to conclude. Further refinement of the indicator, or other indicators may enable you to definitively address these points in the future*.

Millimolar concentrations (1-10 mM) of 8-Br-cGMP and -cAMP are commonly used to stimulate Ca^2+^ entry in mammalian sperm (e.g. [28], Nature; [43]; Biology of Reproduction, Fukuda et al., 2004, Journal of Cell Science). Genetic ablation of CatSper in mice abolishes 8-Br-cNMP-evoked Ca^2+^ entry ([28], Nature), suggesting that these compounds act via CatSper. In [4] (EMBO Journal), we reveal that 8-Br-cGMP in fact directly activates human CatSper, rather than a “CatSper-like” channel, via an extracellular site; the current increases by 100%, which we consider a significant change.

The results in human sperm suggest that these chemicals might also directly activate mouse CatSper. In fact, studies on the action of 8-Br-cNMPs on membrane currents in mouse sperm yielded mixed results: [18] (Nature) did not observe an action of 8-Br-cNMPs (at 5 mM), whereas Cisneros-Mejorado et al. (2014, Andrology, and 2012, FEBS Letters) show that 8-Br-cNMPs (0.2 – 0.8 mM) enhance monovalent currents; these 8-Br-cNMP-induced currents are presumably mediated by CatSper rather than by CNG channels as the authors concluded. We did not scrutinize the direct action of cNMP analogs on mouse sperm and we don’t want to make any unwarranted claims. Therefore, we followed the referee´s suggestion and rephrased the paragraph. We now state that: “in contrast, in wild-type and bPAC sperm, 8-Bromo-cAMP, 8-Bromo-cGMP, and db-cAMP evoked a Ca^2+^ signal (Figure 3) that is abolished in CatSper-null sperm (Figure 3), similar to what has been reported by others ([28]; Xia and Ren, 2007)”.

Reviewer #1:

*The only major comment I have is on the conclusion that cAMP does not induce Ca*^*2+*^
*influx ([Ca*^*2+*^*]i increase) in sperm, which is apparently at odd with various previous studies with “indirect” methods. While the finding in*
Figure 3*, that light stimulation does not lead to detectable change in Δ F/F, is consistent with the idea that cAMP doesn't cause* Ca^2+^
*influx, there can be other interpretations to the data. For example, the light-induced [cAMP] increase is relatively moderate (∼2 to 4 fold) and is comparable to that induced by HCO*_*3*_
*(*Figure 2*). Earlier studies by Babcock and colleagues (e.g. Wennemuth, JBC, 2000) showed that large detectable HCO*_*3*_^*-*^
*induced [Ca*^*2+*^*] increases were obvious only when combined with K8.6 stimulus (higher [K+] with alkali bath). It is possible that at the level achieved by light stimulation in the current study, cAMP requires other co-stimuli to induce Ca*^*2+*^
*influx. Another possibility, perhaps even more likely, is that the 2-4 fold [cAMP] increase, which is measured from whole, is largely from sperm head (because of its overwhelmingly larger volume) but little from sperm principal piece where the Ca*^*2+*^
*influx channels are localized. The experiments in the paper do not show whether there is [cAMP] increase in sperm tail upon light stimulation. The geometry of the sperm midpiece and the filled mitochondria perhaps are significant barriers for cAMP diffusion from sperm head to tail*.

Previously, two different experimental protocols had been applied that prompted different interpretations. First, in mice sperm, a cAMP increase *alone* upon stimulation with HCO_3_^-^ does not stimulate a Ca^2+^ influx (e.g. Wennemuth et al., 2000, Journal of Biological Chemistry; [40], Development; [6], Developmental Biology). These previous results by Dr. Babcock´s group completely agree with our results. However, if Ca^2+^ influx was provoked by high K^+^/alkaline pH (conditions that activate CatSper), HCO_3_^-^ increases and accelerates the amplitude and kinetics, respectively, of the K8.6-stimulated Ca^2+^ increases (e.g. [40], Development). Thus, cAMP might sensitize CatSper to open upon depolarization and alkalization, but cAMP does not open CatSper on its own. Concerning the light-induced cAMP increase in the flagellum, please see our response to the Reviewing Editor.

Reviewer #2:

*This paper describes a novel animal model featuring expression of the light-activated adenylyl cyclase bPAC exclusively in sperm. The authors present exciting data showing rapid changes in flagellar beat in live sperm upon stimulation of bPAC, and that light can serve as a surrogate for other stimuli believed to elevate cAMP in sperm such as HCO*_*3*_^*-*^
*treatment. The changes in flagellar beat also provide a useful (albeit indirect) readout of the temporal features of illumination-dependent cAMP production by bPAC*.

*This optogenetic tool will provide a fantastic opportunity to clarify the role of compartmentalized cAMP signaling in sperm motility, maturation and fertilization. The animal model is well-validated (*Figure 1*), and bPAC expression does not alter expression of other cAMP signaling proteins and pathways (*Figure 2*). While expression of bPAC did not alter PKA phosphorylation per se (*Figure 2*), it would have been nice to see that illumination could also produce sufficient cAMP to stimulate phosphorylation of PKA targets. If the authors could produce such a figure, it would strengthen the paper*.

We performed a full new data set and replaced Figure 2 with a new figure showing both bicarbonate- and light-induced phosphorylation of PKA targets*.* We have amended the Materials and methods section accordingly.

*In*
Figure 3*, the authors confirm previous findings that cyclic nucleotide analogs (10 mM 8Br-cAMP and 10mM db-cAMP) caused Ca*^*2+*^
*entry into the sperm via CatSper, but light stimulation of bPAC did not cause this Ca*^*2+*^
*entry. Is another possible interpretation of this result simply that the illumination of bPAC didn't cause sufficient intracellular cAMP production to influence CatSper? On the other hand, the concentration of cAMP analogs in the present study is much higher than would typically be used in most other cell types to activate cAMP targets. Is this concentration comparable to what was used previously in other studies that examined the effects of cAMP analogs on calcium entry? What was the concentration of db-cAMP used in*
Figure 2
*(cAMP-dependent tyrosine phosphorylation)?*

Concerning the use and action of 8-Br-cNMPs, please see our response to the Reviewing Editor. A db-cAMP-induced Ca^2+^ influx via CatSper has not been reported before. In general, this drug is used at millimolar concentrations to interfere with cAMP signaling. The concentration of db-cAMP used in Figure 2 was 1 mM, similar to what has been used in other studies (Visconti et al., 1955, Development). We have included this information in the figure legend.